# Characterising the nationwide burden and predictors of unkept outpatient appointments in the National Health Service in England: A cohort study using a machine learning approach

Sion Philpott-Morgan[1], Dixa B. Thakrar[2], Joshua Symons[1,2], Daniel Ray[3], Hutan Ashrafian[2]*, Ara Darzi[2]

1 NHS Digital, London, United Kingdom, 2 Institute of Global Health Innovation, Imperial College London, London, United Kingdom, 3 Farr Institute of Health Informatics Research, University College London, London, United Kingdom

* h.ashrafian@imperial.ac.uk

**Data Availability Statement:** The main dataset analysed in this manuscript is called Hospital

## Abstract

### Background

Unkept outpatient hospital appointments cost the National Health Service £1 billion each year. Given the associated costs and morbidity of unkept appointments, this is an issue requiring urgent attention. We aimed to determine rates of unkept outpatient clinic appointments across hospital trusts in the England. In addition, we aimed to examine the predictors of unkept outpatient clinic appointments across specialties at Imperial College Healthcare NHS Trust (ICHT). Our final aim was to train machine learning models to determine the effectiveness of a potential intervention in reducing unkept appointments.

### Methods and findings

UK Hospital Episode Statistics outpatient data from 2016 to 2018 were used for this study. Machine learning models were trained to determine predictors of unkept appointments and their relative importance. These models were gradient boosting machines. In 2017–2018 there were approximately 85 million outpatient appointments, with an unkept appointment rate of 5.7%. Within ICHT, there were almost 1 million appointments, with an unkept appointment rate of 11.2%. Hepatology had the highest rate of unkept appointments (17%), and medical oncology had the lowest (6%). The most important predictors of unkept appointments included the recency (25%) and frequency (13%) of previous unkept appointments and age at appointment (10%). A sensitivity of 0.287 was calculated overall for specialties with at least 10,000 appointments in 2016–2017 (after data cleaning). This suggests that 28.7% of patients who do miss their appointment would be successfully targeted if the top 10% least likely to attend received an intervention. As a result, an intervention targeting the top 10% of likely non-attenders, in the full population of patients, would be able to capture 28.7% of unkept appointments if successful. Study limitations include that some unkept

Episode Statistics (HES) Outpatients. This is not publicly available due to legal restrictions. However, access to this data can be gained by applying to the Data Access request Service at NHS Digital (https://digital.nhs.uk/services/data-access-request-service-dars), providing that the user has good reason and a valid legal basis for accessing it (including a data sharing contract and data sharing agreement), and has satisfied NHS Digital's information governance and data security requirements. Other datasets used for this analysis, such as the Index of Multiple Deprivation, are publicly available (https://www.gov.uk/government/statistics/english-indices-of-deprivation-2019).

**Funding:** This work was funded by the National Institute for Health Research Imperial Biomedical Research Centre (https://imperialbrc.nihr.ac.uk, grant number: 1215-20013 to HA, AD). The funders had no role in study design, data collection and analysis, decision to publish, or preparation of the manuscript.

**Competing interests:** We have read the journal's policy and the authors of this manuscript have the following competing interests: JS currently works for NHS Digital who has produced the models and holds the data used for this publication. SP, DBT, DR, HA and AD have declared that no competing interests exist.

**Abbreviations:** AUROC, area under the receiver operating characteristic curve; GBM, gradient boosting machine; HES, Hospital Episode Statistics; ICHT, Imperial College Healthcare NHS Trust; IMD, Index of Multiple Deprivation; LR, likelihood ratio; NHS, National Health Service; PPV, positive predictive value.

appointments may have been missed from the analysis because recording of unkept appointments is not mandatory in England. Furthermore, results here are based on a single trust in England, hence may not be generalisable to other locations.

## Conclusions

Unkept appointments remain an ongoing concern for healthcare systems internationally. Using machine learning, we can identify those most likely to miss their appointment and implement more targeted interventions to reduce unkept appointment rates.

---

## Author summary

### Why was the study done?

- Unkept appointments cost the National Heath Service £1 billion annually.

### What did the researchers do and find?

- We determined rates of unkept outpatient clinic appointments and examined the predictors of unkept appointments across specialties.

- We found that patients who had previously missed their appointment were most likely to miss another appointment.

### What do these findings mean?

- Based on our results, we have identified a cohort of patients for whom targeted interventions can be implemented, to reduce the number of unkept appointments in order to improve patient care and reduce costs to healthcare providers.

## Background

Unkept hospital appointments, also known as "did not attends" (DNAs), are a dilemma facing multiple healthcare systems worldwide. In 2017–2018, 8 million National Health Service (NHS) hospital appointments, almost 1 in 10, were unkept in England. Each outpatient hospital appointment is estimated to cost the NHS £120, yielding an overall cost for the system of approximately £1 billion in unkept appointments [1–3]. It is estimated that unkept appointments cost the healthcare system in the US $150 billion a year [4]. The financial and public health impacts of unkept appointments are therefore vast. It is wasteful of resources and may also increase patient morbidity and lengthen waiting lists from 1 week to up to 6 months [5,6]. A nationwide study in Scotland reported that those who missed more than 2 appointments had a 3-fold increase in hazards of mortality compared to those who did not miss

appointments [6]. The NHS is a service with limited resources and is ever under the stress of financial limitations; hence, unkept appointments need to be addressed to ensure resources are allocated appropriately.

Reasons for unkept appointments often include forgetfulness, being unaware of the appointment, feeling too unwell to attend, hospital administrative errors, work commitments, transport difficulties, and resolution of symptoms [7–20].

In the absence of a more predictive and proactive approach to mitigate unkept appointments, outpatient clinics often overbook appointments. This puts constraints on those delivering clinic care. Other proactive interventions such as appointment reminders by phone call, letter, or text message have been implemented to try to reduce the number of unkept appointments. Other strategies have included giving patients the responsibility of booking their appointment, either through Freephone service or online [5]. Interventions may fail as they are often targeted using a blanket approach, without knowing which cohort of patients or clinical factors are most effective to target. Interventions need to be targeted effectively to increase patient engagement and therefore aid the ongoing issue of tackling health inequalities.

In order to understand how best to target interventions to reduce the number of unkept outpatient clinic appointments, it is first important to understand the predictors and characteristics of unkept appointments. Previous studies have employed machine learning and statistical modelling to predict unkept appointments [21–24]. However, such modelling has not been carried out in a UK setting across all medical specialties, but rather has been limited to individual specialties or restricted to primary care or community settings.

Our aim was to demonstrate the rates of unkept outpatient clinic appointments across hospital trusts in England, with an added breakdown by specialty. To appraise the potential utility of this approach at a local level, we examined the predictors of unkept outpatient clinic appointments across specialties at a single trust within England, Imperial College Healthcare NHS Trust (ICHT). Finally, we trained machine learning models to determine the effectiveness of a potential intervention in reducing unkept appointments.

## Methods

### Data capture

We present a data flow diagram in Fig 1. We used Hospital Episode Statistics (HES) outpatient data spanning England from April 2016 to March 2018 to generate and train the models used for this study. Data codings and definitions can be found on the NHS Digital website [2]. HES is a database detailing all admissions and emergency and outpatient appointments at NHS hospitals. HES provides a number of patient characteristics including age, sex, ethnicity, and geographical information. Index of Multiple Deprivation (IMD) demographic data were also used—a dataset that is openly available [25].

### Data cleaning

The submission of kept outpatient appointment data to NHS Digital is mandatory in England. However, the submission of unkept appointments is not mandatory. Some sites have a reported 0% unkept appointment rate in the data they submit. We excluded these sites from this analysis, along with the bottom and top 10% of outliers.

Our goal was to predict outpatient non-attendance without warning. Hence, appointments that were cancelled in advance, either by patients or consultants, were also excluded from this analysis.

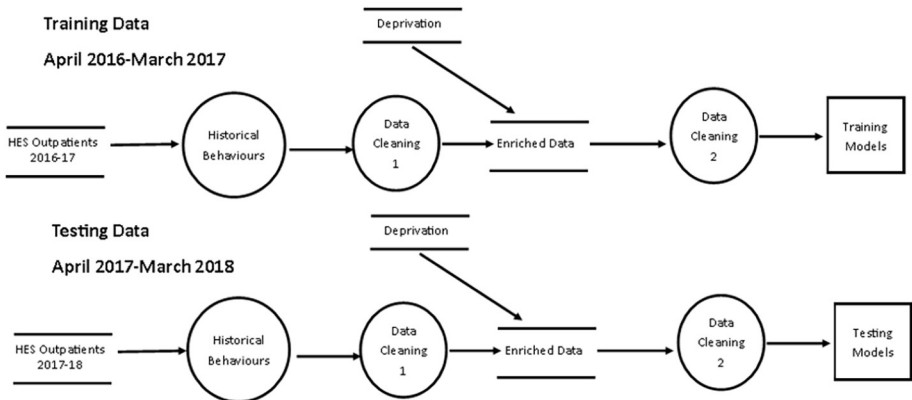

1. **Historical Behaviours**. Calculating number of appointments, unkept appointments and cancellations over the previous 12 months, then adding to the data as additional columns. Also, formatting features.
2. **Data Cleaning Step 1**. Removing small sites (<500 appointments annually) and sites with 0 DNAs.
3. **Enriched data**. Joining HES Outpatients and IMD deprivation deciles.
4. **Data Cleaning Step 2**. Removing records with missing values.
5. **Training/ Testing models using the processed data**. Models trained using 2016-2017 data, then tested using 2017-18 data.

**Fig 1. Data flow diagram.** Numbers of appointments and unkept appointments were obtained for the top 50 trusts by appointment volume. Unkept appointments within Imperial College Healthcare NHS Trust were further broken down by specialty. DNA, did not attend; HES, Hospital Episode Statistics; IMD, Index of Multiple Deprivation.

Appointments from April 2016 to March 2017 were used to train the models. The recency (in days) and frequency (over the previous 12 months) of appointments and unkept appointments were calculated and included as predictors (see S1 Text for details).

We tested the models using ICHT-specific data from April 2017 to March 2018. This again included the number of appointments, unkept appointments, and cancellations, calculated by specialty, from the previous 12 months.

## Statistical analysis

The recorded outcome variable was binary, indicating whether the patient attended their appointment or did not attend their appointment without providing advance warning. Note however that the output of the models was a probability between 0 and 1, indicating the likelihood of a given patient not showing up to an appointment prior to the fact. Advance cancellations were not included in the data.

Variables used in the model were defined as shown in Table 1.

Using the R statistical programming language, we trained machine learning models for each of the top 100 treatment specialties by national volume in 2016–2017 to determine predictors of unkept appointments and their relative importance (S1 Text). These models were gradient boosting machines (GBMs). HES data from 2016–2017 were used to train the models.

The models were then tested using 2017–2018 HES outpatient data. We conducted an analysis based on a hypothetical intervention targeting the top 10% of outpatient appointments by risk. The intervention here could be a phone call reminder or a virtual consultation. This was a post hoc analysis, based on observational data, and did not have a prespecified study plan.

**Test metrics.** Model sensitivity is the proportion of unkept appointments captured by the intervention. For example, a sensitivity of 0.33 implies that 33% of unkept appointments are captured; that is, the intervention could at best reduce unkept appointments by 33%. The

**Table 1. Predictor variable definition.**

| Predictor variable | Description |
|---|---|
| Unkept appointments last 12 months | Count of number of unkept appointments in last 12 months |
| Appointments last 12 months | Count of number of outpatient appointments in last 12 months |
| Cancellations last 12 months | Count of number of cancellations in last 12 months |
| Days since last unkept appointment | Days since the previous unkept appointment |
| Days since last appointment | Days since the previous outpatient appointment (kept or unkept) |
| Days since last cancellation | Days since the previous cancellation |
| Lead care professional | Patient to be seen by the lead care professional versus another member of the professional team |
| APPTAGE CALC | Age at appointment—babies under 1 year decimalised |
| REFSOURC | Source of referral |
| Health deprivation score | Health Deprivation and Disability subscale from Index of Multiple Deprivation (IMD) |
| IDAOPI score | Income Deprivation Affecting Older People Index subscale from IMD |
| IDACI score | Income Deprivation Affecting Children Index subscale from IMD |
| IMD score | Overall Index of Multiple Deprivation Scale |
| Sex | Sex of patient |
| Weekday | Day of week (Monday, Tuesday, Wednesday, etc.) |
| Consultation | Service type requested |
| Appointment type | First attendance, follow-up, or telephone |

positive predictive value (PPV) is the proportion of the time the model is right. For example, suppose that 100 people are chosen for the intervention. If the PPV is 0.5, then this tells us that 50 out of 100 would have missed their appointment if there was no intervention. The likelihood ratio (LR) tells us how much more likely those who are selected for an intervention are to have an unkept appointment, in comparison to those who are not selected. The area under the receiver operating characteristic curve (AUROC) tells us how good the model is at distinguishing classes, in this instance between patients who will attend their appointment versus those who will not.

**Prediction metrics.** The prediction metrics included the percentage importance of the different predictors for a given speciality (defined in terms of reduction in predictive error), the average importance for a predictor across all outpatient specialties, and the variation in importance for a predictor across all outpatient specialties.

See S1 Table (RECORD Checklist) for details on our reporting in this study.

As per the Health and Social Care Act 261 and the Data Protection Act 2018, as a national institution, NHS Digital is directed to store and analyse secondary care data. Internal approval for this project was granted by the information asset owner.

# Results

## Across the whole NHS

In both 2016–2017 and 2017–2018, there were approximately 97–98 million outpatient appointments after removing advance cancellations, and data cleaning. The rate of unkept appointments in these datasets was 8.0% and 8.1%, respectively.

Across trusts in England, focusing on the top 50 trusts by appointment volume, the rate of unkept appointments ranged from 3.9% to 14.8%. Fig 2 shows the top 20 trusts with the

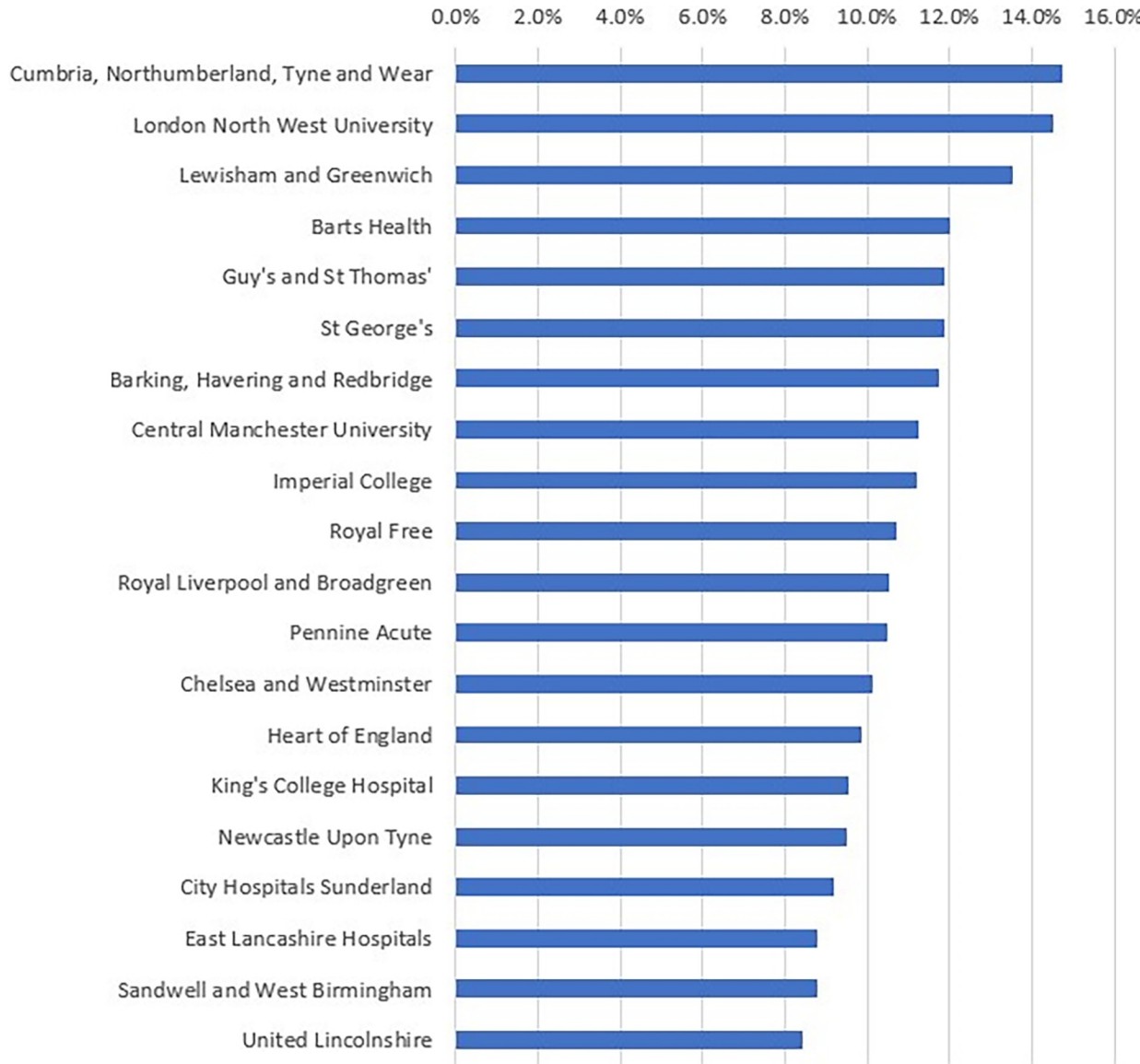

**Fig 2. Unkept appointment rates by UK hospital trust (excluding advance cancellations).**

highest unkept appointment rates. The full table can be found in S2 Table. Eight of the 10 trusts with the highest unkept appointment rates were within the London region.

## Within Imperial College Healthcare NHS Trust

Focusing on ICHT only, there were approximately 1.1 million outpatient appointments, excluding advance cancellations and sites with very low volumes. Approximately 910,000 remained after data cleaning. The rate of unkept appointments in this data set was 11.2%. The rates of the highest and lowest 5 specialties are presented in Table 2. Hepatology had the highest rate of unkept appointments (17%), and medical oncology had the lowest (6%). See S3 Table for all specialties.

**Table 2. Highest and lowest unkept appointment rates by speciality.**

| Speciality name | Number of appointments | Number of unkept appointments | Unkept appointment rate (%) |
|---|---|---|---|
| **Highest rates of unkept appointments** | | | |
| Hepatology | 14,731 | 2,494 | 17% |
| Diabetic medicine | 12,182 | 1,929 | 16% |
| Ophthalmology | 74,442 | 11,188 | 15% |
| Ear, nose, and throat | 34,820 | 5,109 | 15% |
| Vascular surgery | 12,666 | 1,772 | 14% |
| **Lowest rates of unkept appointments** | | | |
| Gynaecology | 62,526 | 5,508 | 9% |
| Breast surgery | 16,838 | 1,428 | 8% |
| Anaesthetics | 15,690 | 1,180 | 8% |
| Audiological medicine | 20,577 | 1,647 | 8% |
| Medical oncology | 36,466 | 2,052 | 6% |

**Predictors of outpatient non-attendance overall.** Predictor importance, sorted by average importance, for our composite prediction model across all specialties is shown in Table 3. Days since the previous unkept appointment (recency) was responsible on average for 25%, and the number of unkept appointments a patient had in the previous 12 months (frequency) was responsible on average for 13% of the predictive value of these models. Age at time of appointment accounted for around 10% of the predictive value. A larger number of previous unkept appointments was associated with an increased risk of failing to keep a future appointment. A larger number of previous cancellations was associated with an increased risk of failing to keep a future appointment. Older patients were least likely to miss an appointment. More deprived areas (lower IMD decile) were associated with an increased risk of an unkept

**Table 3. Overall predictor importance.**

| Predictor | Mean | Rank average | Variation |
|---|---|---|---|
| Days since last unkept appointment | 25% | 1.2 | 0.2 |
| Unkept appointments last 12 months | 13% | 3.0 | 1.7 |
| Age at appointment | 10% | 4.6 | 8.8 |
| Lead care professional | 9% | 5.1 | 6.5 |
| Appointments last 12 months | 7% | 5.3 | 4.7 |
| Days since last appointment | 6% | 6.0 | 4.2 |
| Referral source | 5% | 7.1 | 8.2 |
| Health deprivation score | 5% | 7.1 | 4.0 |
| IDAOPl score | 5% | 7.7 | 7.6 |
| IDACI score | 3% | 10.8 | 3.8 |
| IMD score | 2% | 12.0 | 4.4 |
| Weekday | 2% | 11.7 | 3.6 |
| Appointment type | 2% | 12.8 | 5.9 |
| Days since last cancellation | 2% | 12.7 | 3.1 |
| Consultation | 2% | 13.9 | 5.4 |
| Cancellations last 12 months | 1% | 15.6 | 2.5 |
| Sex | 0% | 16.5 | 0.8 |

IDACI, Income Deprivation Affecting Children Index; IDAOPI, Income Deprivation Affecting Older People Index; IMD, Index of Multiple Deprivation.

appointment. Seeing a lead care professional was associated with a decreased risk of an unkept appointment.

**Predictors of outpatient non-attendance by specialty.** Fairly consistently across the specialties with the highest and lowest unkept appointment rates, days since the previous unkept appointment and the number of previous unkept appointments in the last 12 months were among the most important predictors (Tables 4 and 5). The full table for all specialties can be seen in S4 Table. Age at time of appointment and number of appointments in the last 12 months were also important. Age at appointment was the most variable in terms of its importance for a given specialty. For instance, age at appointment is relatively important as a predictor of attendance for audiological medicine; ear, nose, and throat; and ophthalmology, but relatively unimportant for hepatology and vascular surgery. Referral source also varied in importance by specialty. Number of days since the last unkept appointment was consistently among the top 2 predictors, while sex was consistently among the bottom 4.

### GBM model

In order to predict rates of unkept appointments, GBM models were trained on the top 100 specialties by appointment volume. Data from 2016–2017 were used to train the models, and data from 2017–2018 were used to test them. The test metrics for the specialties with the highest and lowest unkept appointment rates are shown in Table 6. From these models, we calculated the sensitivity, LR, and PPV for generating interventions in selected proportions of non-attenders to assess potential clinical improvements in attendance.

**Sensitivity at 10% cutoff.** A sensitivity of 0.28 for hepatology suggests that 28% of patients who do miss their appointment would be successfully targeted if the top 10% least likely to attend received an intervention. As a result, an intervention targeting the top 10% of likely non-attenders, in the full population of patients, would be able to capture 28% of unkept appointments if successful.

**Likelihood ratio.** The LR for the top 5 specialties was greater than 3, meaning those patients selected by the models for an intervention were at least 3 times as likely to miss their appointment than those who were not selected for an intervention.

**Positive predictive value.** The PPV for the top 5 specialties was between 37% and 47%, meaning that of those selected for the targeted intervention, 37%–47% would be expected to miss their appointment prior to the intervention. This is in comparison to a 14%–17% unkept appointment rate across all appointments for the top 5 specialties.

Among the bottom 5 specialties, of those selected for an intervention roughly 16%–29% would be expected to miss their appointment, in comparison to a 6%–8% unkept appointment rate across all appointments in the bottom 5 specialties.

**Area under the curve.** As a metric of model performance, the AUROC was fairly consistent across both the top 5 and bottom 5 specialties, ranging from 0.67 to 0.74.

So long as the cost of an intervention is less than one-third of the average cost of the potential reduction in unkept appointments, then using these models for targeted interventions would theoretically be cost-effective.

## Discussion

Unkept appointments are a worldwide issue, causing inequalities in health and inefficient use of resources. Using a national data-driven approach, we determined national and local unkept outpatient appointment rates across secondary care in the United Kingdom, and across multiple specialties at a single hospital trust. Previous nationwide studies have looked primarily at general practice data [6]. In our study, rate of unkept appointments varied across NHS hospital

**Table 4. Predictor rank for each specialty with the highest and lowest unkept appointment rates: Mean percentage.**

| Speciality name | Appointments last 12 months | Unkept appointments last 12 months | Cancellations last 12 months | Days since last appointment | Days since last unkept appointment | Days since last cancellation | Age at appointment | IDAOPI score | IDACI score | IMD score | Health deprivation score | Consultation | Lead care professional | Appointment type | Sex | Referral source | Weekday |
|---|---|---|---|---|---|---|---|---|---|---|---|---|---|---|---|---|---|
| **Highest rates of unkept appointments** | | | | | | | | | | | | | | | | | |
| Hepatology | 9.3% | 16.4% | 0.6% | 7.8% | 29.5% | 1.7% | 5.9% | 2.4% | 2.3% | 4.3% | 4.8% | 3.7% | 3.7% | 0.9% | 0.4% | 3.3% | 3.1% |
| Diabetic medicine | 12.3% | 15.1% | 0.6% | 9.5% | 19.3% | 1.4% | 6.7% | 1.5% | 1.3% | 0.9% | 5.2% | 0.8% | 14.8% | 4.5% | 0.9% | 4.0% | 1.3% |
| Ophthalmology | 5.9% | 11.4% | 0.5% | 4.3% | 28.3% | 1.1% | 13.6% | 6.4% | 3.9% | 3.1% | 4.8% | 1.1% | 6.3% | 0.7% | 0.3% | 6.9% | 1.4% |
| Ear, nose, and throat | 3.8% | 9.5% | 0.4% | 4.4% | 25.4% | 1.6% | 15.2% | 5.0% | 2.6% | 1.6% | 7.2% | 0.8% | 16.0% | 1.4% | 0.4% | 3.3% | 1.6% |
| Vascular surgery | 5.6% | 14.3% | 1.2% | 4.9% | 25.9% | 1.1% | 5.4% | 3.5% | 1.2% | 1.3% | 5.7% | 2.6% | 20.1% | 1.6% | 0.4% | 3.4% | 1.7% |
| **Lowest rates of unkept appointments** | | | | | | | | | | | | | | | | | |
| Gynaecology | 3.4% | 7.8% | 0.6% | 8.5% | 26.9% | 2.4% | 6.1% | 4.4% | 2.7% | 2.2% | 7.6% | 1.5% | 11.4% | 3.7% | 0.0% | 8.2% | 2.5% |
| Breast surgery | 5.7% | 7.5% | 0.3% | 7.0% | 37.1% | 1.4% | 10.6% | 4.3% | 3.0% | 1.8% | 4.9% | 1.4% | 6.5% | 4.0% | 0.0% | 2.8% | 1.6% |
| Anaesthetics | 4.5% | 7.6% | 0.7% | 5.3% | 34.8% | 2.3% | 14.0% | 6.1% | 2.5% | 1.8% | 3.2% | 0.2% | 3.8% | 2.8% | 0.8% | 6.5% | 3.3% |
| Audiological medicine | 2.9% | 11.1% | 0.6% | 2.3% | 13.2% | 1.4% | 29.0% | 11.9% | 3.5% | 2.5% | 4.3% | 0.3% | 7.3% | 0.5% | 0.3% | 7.7% | 1.3% |
| Medical oncology | 7.2% | 8.2% | 1.6% | 8.2% | 13.7% | 3.6% | 6.3% | 10.5% | 6.9% | 6.4% | 8.3% | 5.3% | 5.3% | 1.8% | 0.4% | 3.3% | 3.2% |

IDACI, Income Deprivation Affecting Children Index; IDAOPI, Income Deprivation Affecting Older People Index; IMD, Index of Multiple Deprivation.

**Table 5. Predictor rank for each specialty with highest and lowest unkept appointment rates: Rank average.**

| Speciality name | Appointments last 12 months | Unkept appointments last 12 months | Cancellations last 12 months | Days since last appointment | Days since last unkept appointment | Days since last cancellation | Age at appointment | IDAOPI score | IDACI score | IMD score | Health deprivation score | Consultation | Lead care professional | Appointment type | Sex | Referral source | Weekday |
|---|---|---|---|---|---|---|---|---|---|---|---|---|---|---|---|---|---|
| **Highest rates of unkept appointments** | | | | | | | | | | | | | | | | | |
| Hepatology | 3 | 2 | 16 | 4 | 1 | 14 | 5 | 12 | 13 | 7 | 6 | 8 | 9 | 15 | 17 | 10 | 11 |
| Diabetic medicine | 4 | 2 | 17 | 5 | 1 | 11 | 6 | 10 | 12 | 15 | 7 | 16 | 3 | 8 | 14 | 9 | 13 |
| Ophthalmology | 7 | 3 | 16 | 9 | 1 | 13 | 2 | 5 | 10 | 11 | 8 | 14 | 6 | 15 | 17 | 4 | 12 |
| Ear, nose, and throat | 8 | 4 | 17 | 7 | 1 | 12 | 3 | 6 | 10 | 13 | 5 | 15 | 2 | 14 | 16 | 9 | 11 |
| Vascular surgery | 5 | 3 | 15 | 7 | 1 | 16 | 6 | 8 | 14 | 13 | 4 | 10 | 2 | 12 | 17 | 9 | 11 |
| **Lowest rates of unkept appointments** | | | | | | | | | | | | | | | | | |
| Gynaecology | 10 | 5 | 16 | 3 | 1 | 13 | 7 | 8 | 11 | 14 | 6 | 15 | 2 | 9 | 17 | 4 | 12 |
| Breast surgery | 6 | 3 | 16 | 4 | 1 | 14 | 2 | 8 | 10 | 12 | 7 | 15 | 5 | 9 | 17 | 11 | 13 |
| Anaesthetics | 7 | 3 | 16 | 6 | 1 | 13 | 2 | 5 | 12 | 14 | 10 | 17 | 8 | 11 | 15 | 4 | 9 |
| Audiological medicine | 9 | 4 | 14 | 11 | 2 | 12 | 1 | 3 | 8 | 10 | 7 | 17 | 6 | 15 | 16 | 5 | 13 |
| Medical oncology | 6 | 4 | 16 | 5 | 1 | 12 | 9 | 2 | 7 | 8 | 3 | 11 | 10 | 15 | 17 | 13 | 14 |

Green corresponds to the most important predictors, and red to the least important. IDACI, Income Deprivation Affecting Children Index; IDAOPI, Income Deprivation Affecting Older People Index; IMD, Index of Multiple Deprivation.

**Table 6. Gradient boosting machine validation metrics of specialties with the highest and lowest unkept appointment rates.**

| Speciality name | Number of appointments | Number of unkept appointments | Unkept appointment percent | Sensitivity | PPV | LR | AUROC |
|---|---|---|---|---|---|---|---|
| **Highest rates of unkept appointments** | | | | | | | |
| Hepatology | 14,731 | 2,494 | 17% | 0.28 | 0.46 | 4.10 | 0.74 |
| Diabetic medicine | 12,182 | 1,929 | 16% | 0.29 | 0.47 | 4.62 | 0.74 |
| Ophthalmology | 74,442 | 11,188 | 15% | 0.26 | 0.39 | 3.63 | 0.71 |
| Ear, nose, and throat | 34,820 | 5,109 | 15% | 0.26 | 0.38 | 3.61 | 0.69 |
| Trauma and orthopaedics | 44,966 | 6,460 | 14% | 0.25 | 0.37 | 3.45 | 0.69 |
| **Lowest rates of unkept appointments** | | | | | | | |
| Gynaecology | 62,526 | 5,508 | 9% | 0.31 | 0.27 | 3.82 | 0.72 |
| Breast surgery | 16,838 | 1,428 | 8% | 0.35 | 0.29 | 4.48 | 0.72 |
| Audiological medicine | 20,577 | 1,647 | 8% | 0.24 | 0.19 | 2.67 | 0.67 |
| Anaesthetics | 15,690 | 1,180 | 8% | 0.26 | 0.20 | 2.99 | 0.67 |
| Medical oncology | 36,466 | 2,052 | 6% | 0.29 | 0.16 | 3.30 | 0.69 |

AUROC, area under the receiver operating characteristic curve; LR, likelihood ratio; PPV, positive predictive value.

trusts from 4.3% to 15.1%. The higher rates seen in London may be explained by the more heterogenous population within London, with language barriers, transport failures, and administrative failures.

Predictors of unkept appointments can be divided into clinical, behavioural, and sociodemographic. At ICHT, there was great heterogeneity in unkept appointment rates across all specialties. The highest rates were seen in hepatology; diabetes; ophthalmology; ear, nose, and throat; and vascular surgery patients. Interestingly, these patients' co-morbidities may overlap. For example, a patient with diabetes may require ophthalmology review of diabetic retinopathy or vascular review for diabetic foot ulcers. It is not clear why hepatology had the highest rate of unkept appointments; however, studies looking at gastroenterology patients found that causes of unkept appointments included forgetting their appointment or clerical errors [10]. A cohort study of 521 unkept ophthalmology appointments found that the top reasons for non-attendance were not feeling well enough to attend, forgetting the appointment, administrative errors, and that their condition had improved [26].

When stratifying by the predictors of unkept appointments, there were similarities and differences across all specialties. Consistently, having a prior unkept appointment was the greatest predictor across all specialties except medical oncology. This suggests that behaviour is the most important predictor, and hence behaviour-related interventions targeting those with recurrent unkept appointments is necessary. Therefore, adopting a targeted approach to reducing unkept appointments maybe more effective than a blanket approach.

Sex was the least important predictor for the majority of specialties. This contradicts other findings in the literature. Previous studies suggested that sex was a predictor of non-attendance, with males having a higher risk [6,12,17,27]. Similarly, deprivation did not rank very highly, in contrast to existing literature [11,13,28–30]. A possible reason for this is that the models generated here had greater granularity and included predictors of greater importance that could not always be captured in other studies.

Gynaecology, breast surgery, anaesthetics, audiological medicine, and medical oncology had the lowest rates of unkept appointments. Oncology patients may have better adherence to treatment due to the mortality associated with their disease and hence may be more likely to attend their appointments. Whilst not all, a significant proportion of gynaecological and breast patients may also fall under oncology.

Targeted interventions could be implemented at multiple levels: organisational, psychosocial, or through information dissemination. For example, virtual clinics could be a practical solution, and have been trialled across multiple specialties [31–33]. Other interventions include stating the cost of the appointment when sending SMS reminders to patients, which has been shown in a trial to reduce unkept appointment rates compared to SMS reminders not stating appointment costs [34]. Shared appointments, where patients receive consultations with their doctor in the presence of other patients with similar conditions, may provide another means of reducing unkept appointment rates [35]. Such interventions would have to be trialled, and an assessment of utility, safety, cost-effectiveness, and patient satisfaction would have to be undertaken.

Whilst text message reminders have been shown to reduce unkept appointment rates, patients are still missing their appointments, and hence the present findings will allow us to go a step further by introducing targeted interventions. In addition, vulnerable, elderly, and deprived patients may not have access to a mobile phone, and therefore would not benefit from a blanket approach using SMS reminders. They may also be most at harm, should they miss their appointment, which again calls for a more targeted approach to ensure they receive the appropriate care.

Aside from the cost implications of unkept appointments, there is increased mortality associated with missing appointments, as seen in general practice. A nationwide study in Scotland reported that those who missed more than 2 appointments had a 3-fold increase in hazards of mortality compared to those who did not miss appointments [6]. Such data are lacking in the secondary care setting.

The GBM models output unkept appointment propensity scores, helping us rank patients in order of which patients are most likely to miss their appointments. The idea is to target a certain proportion of patients, implement an intervention, and decrease unkept appointment rates with minimal effort, rather than targeting all patients. In this way, interventions can be introduced more cost-effectively. Using the 5 specialties with the highest rates of unkept appointments, the models suggest that so long as the cost of an intervention is less than one-third of the average cost of the potential reduction in unkept appointments, using these models for targeted interventions would theoretically be cost-effective. In the context of analysing many specialties, we used a uniform sensitivity cutoff of 10%. In a live service, we could have different cutoffs for different specialties, based on their resources and respective rates of unkept appointments.

Overall, the top 2 predictors—namely, recency and frequency of previous unkept appointments—accounted for 38% of the average predictive value. This highlighted how a simple intervention based on these 2 predictors might have some utility. But it also highlights one of the advantages of applying machine learning models to predictive problems such as this—namely, that the contribution of many factors, along with the complexities of their interactions, can be accounted for in a way that focusing on just a few key factors does not allow.

It is likely that in order to run and implement these models and then apply targeted interventions across the population, technology, and possibly artificial intelligence, will be utilised. In February 2019, the Topol review was published [36]. This review, commissioned by the UK secretary of health, was designed to elucidate how the NHS can make the most of technology

to improve services and help ensure their sustainability [37]. Digital medicine and artificial intelligence can aid in decision-making processes such as booking systems and targeting interventions, as well as utilising the vast volume of data available to generate the models.

As with any study relying on the use of routinely collected data, there are a number of limitations. The submission of kept outpatient appointment data to NHS Digital is mandatory in England. However, the submission of unkept outpatient appointment data is not. Not all trusts report unkept appointments consistently, so the data here may not reflect the true unkept appointment rate. This too may be the case within specialties at a single trust. Hospitals reporting a 0% rate of unkept appointments were excluded from this analysis, along with the bottom and top 10% of outliers. Data with the most missingness would have been among the top 10%. In addition, the results by specialty here are based on a single trust in England; hence, the generalisability of the results across the whole country or other countries may be questioned. Furthermore, our data did not include mental health services, as mental health data are found in the Mental Health Services Data Set (MHSDS) rather than HES. In addition, ICHT does not have a dedicated psychiatry department and utilises liaison psychiatry services from partner London trusts.

We excluded the 10% of trusts with the highest rates of unkept appointments. Including them would have potentially resulted in the model underpredicting unkept appointments. However, we also excluded the 10% of outliers with the lowest rates of unkept appointments, and so there is likely to be some offsetting. Additionally, excluding records that were likely to have lower data quality would have improved the accuracy of the predictions. However, as a limitation, trusts thus excluded would be less represented in the data, so the model would again be less generalisable. Furthermore, in our analysis, we excluded cancellations as we did not have cancellation dates. This may have affected the applicability of the model. Appointments can be cancelled by patients or by the care provider, and cancellations may occur shortly before an appointment, or well in advance. In an ideal scenario, we would know when an intervention for reducing unkept appointments was applied, and filter cancellations accordingly. For example, if patients are called 3 days before an appointment, then we could include cancellations within 3 days of appointments as unkept appointments, while excluding all cancellations that had happened before the phone call reminder.

Nonetheless, the study highlights the importance of repeating such an exercise across other datasets. We would recommend digital health policy makers mandate trusts to record and submit unkept appointments, in addition to those attended, to avoid this issue for future related research.

This study has identified the prevalence of unkept appointments nationally, by trust and broken down by specialty within a single UK trust. The clinical implications are that those locations and specialties with the highest rates may require intervention. The granularity of the predictors allows us to identify which patients are best targeted to implement such interventions, to ensure a reduction in unkept appointments, to ultimately reduce the morbidity associated with them and the waste of resources. Further study is needed in other UK trusts and in other countries to better understand this issue globally, so as to tackle healthcare inequalities. Understanding the complications of unkept appointments in a secondary care setting would also be pertinent as it may aid in clinical decision making and follow-up planning.

The new methods of modelling unkept appointments introduced in this study allow us to have a deeper understanding of the root causes of unkept appointments at the national and local level and may offer a path to offer novel interventions in order to address these causes. Future work should break down these unkept appointments and their causes into clinical, behavioural, and psychosocial domains so that specific targets in these areas can be generated to minimise unkept appointment loads. These approaches will need validation with other

datasets and in formalised clinical trial settings to address the global issue of unkept appointments at the national and local level. The lessons derived from these approaches to unkept appointments may therefore in turn be a route to increase efficacy and efficiency in an era of healthcare rationing and financial constraint.

## Supporting information

**S1 Table. RECORD checklist.**
(DOCX)

**S2 Table. Unkept appointment rates by UK hospital trust.**
(DOCX)

**S3 Table. Unkept appointment rates and model metrics for ICHT in 2017–2018 by specialty.**
(DOCX)

**S4 Table. Predictor importance for each specialty.** Filtered for specialties with at least 10,000 appointments in 2016–2017 (after data cleaning).
(DOCX)

**S1 Text. Model specification.** Fig A: Risk ratio of non-attendance for patients with an unkept appointment in the past 12 months.
(DOCX)

## Author Contributions

**Conceptualization:** Sion Philpott-Morgan, Daniel Ray, Hutan Ashrafian, Ara Darzi.

**Supervision:** Hutan Ashrafian, Ara Darzi.

**Writing – original draft:** Sion Philpott-Morgan, Dixa B. Thakrar.

**Writing – review & editing:** Sion Philpott-Morgan, Dixa B. Thakrar, Joshua Symons, Daniel Ray, Hutan Ashrafian.

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
