## [Editor Report · Decision Letter 0]

24 Feb 2020

Dear Dr Thakrar, 

Thank you for submitting your manuscript entitled "Characterising and reducing the burden of unkept outpatient appointments in the NHS through a national data-driven machine learning approach" for consideration by PLOS Medicine.

Your manuscript has now been evaluated by the PLOS Medicine editorial staff and I am writing to let you know that we would like to send your submission out for external peer review.

Kind regards,

Helen Howard, for Clare Stone PhD 

Acting Editor-in-Chief

PLOS Medicine 

plosmedicine.org

---

## [Decision Letter · Decision Letter 1]

8 Dec 2020

Dear Dr. Thakrar,

Thank you very much for submitting your manuscript "Characterising and reducing the burden of unkept outpatient appointments in the NHS through a national data-driven machine learning approach" (PMEDICINE-D-20-00544R1) for consideration at PLOS Medicine. 

Your paper was evaluated by a senior editor and discussed among all the editors here. It was also discussed with an academic editor with relevant expertise, and sent to three independent reviewers; their comments are under the editors' signoff and via the link below, and I hope you find them constructive.

[LINK]

In light of these reviews, I am afraid that we will not be able to accept the manuscript for publication in the journal in its current form, but we would like to consider a revised version that addresses the reviewers' and editors' comments. Obviously we cannot make any decision about publication until we have seen the revised manuscript and your response, and we plan to seek re-review by one or more of the reviewers. 

We expect to receive your revised manuscript by Dec 29 2020 11:59PM. Please email us (plosmedicine@plos.org) if you have any questions or concerns.

We look forward to receiving your revised manuscript. 

Sincerely,

Emma Veitch, PhD

PLOS Medicine

On behalf of Richard Turner, PhD, Senior Editor, 

PLOS Medicine

plosmedicine.org

*Please structure your abstract using the PLOS Medicine headings (Background, Methods and Findings, Conclusions). We'd suggest including in the last sentence of the Abstract Methods and Findings section a note about any key limitation(s) of the study's methodology. Potentially this might include noting (this factor could also be included in the main Discussion section if the authors agree it is relevant) that independent validation in a separate dataset was not done for the predictors that emerged from the current machine learning study.

*At this stage, we ask that you include a short, non-technical Author Summary of your research to make findings accessible to a wide audience that includes both scientists and non-scientists. The Author Summary should immediately follow the Abstract in your revised manuscript. This text is subject to editorial change and should be distinct from the scientific abstract. Please see our author guidelines for more information: https://journals.plos.org/plosmedicine/s/revising-your-manuscript#loc-author-summary

*Please reformat the citation style into PLOS Medicine's format (should be straight forward if using referencing software) - this should use callouts formatted as sequential numerals in square brackets (not superscript).

*Please clarify if the analytical approach followed here corresponded to one laid out in a prospectively developed protocol or analysis plan? We'd ask the authors state this (either way) early in the Methods section.

*The editors would suggest referring to an appropriate reporting guideline to support reporting of the study, and one that may be appropriate is the RECORD guideline (https://www.equator-network.org/reporting-guidelines/record/), developed for reporting of observational routinely-collected data. If the authors agree this is appropriate please enclose the completed RECORD checklist as supporting information with the revised paper.

*We'd suggest including a descriptor of the study design as part of the manuscript title (normally this would be in the subtitle, after a colon - with the first part of the title including the study question/objective). 

*In the Introduction, the authors summarise prior evidence (in the section beginning "Reasons for unkept appointments often include...") but it would be good to then set out the unanswered question driving the current analysis, ie give the reader some idea of the uncertainty around prior research on unkept appointments and therefore what remains to be understood. 

Comments from the reviewers:

Reviewer #1:

"Characterising and reducing the burden of unkept outpatient appointments in the NHS through a national data-driven machine learning approach" studies the rate of outpatient clinic appointments within the National Health Service (NHS) of England, and employs machine learning models (in particular, gradient boosting machines; GBMs) to predict future missed appointments ("Did Not Attends"; DNAs), from available predictor variables (e.g. past appointments, unkept/cancelled appointments, etc - see Table 1 & Appendix 3). Accurate prediction of DNAs holds out the promise of substantial cost savings, since each (missed) appointment costs £120 to the NHS, for an annual wastage of about £1 billion.

Overall, this study raises the potential for both improving cost savings and patient health outcomes, through relatively easily-implemented means. The scale and scope of the HES (Hospital Episode Statistics) Outpatient Data used (containing close to 100 million appointments, and over 100 treatment specialties) is also a particular strength of the evaluation. However, there appear to remain a number of fairly significant issues with the current presentation, that might be addressed:

1. The first two citations supporting the costs of missed appointments to the NHS and the US healthcare system are to webpages. Might there be any alternative peer-reviewed publications that support these figures?

2. The background discussion includes prior work describing prevalence of and reasons for missing appointments, but does not really cover the closest relevant topic (i.e. on using machine learning/statistical methods to predict future unkept appointments) in detail. There appears to be a fair number of such papers, e.g. "Deprivation, demography and missed scheduled appointments at an NHS primary dental care and training service", West et al., British Dental Journal 228 (98-102), 2020; "Modeling patient no-show history and predicting future outpatient appointment behavior in the Veterans Health Administration", Goffman et al., Military Medicine 182 (5-6), e1708-e1714, 2017, etc.

3. The major methodological concern would be with the separation between the training dataset and the validation dataset. In particular, it is stated that "Appointments from April 2016 to March 2017 were used to train the models. The number of appointments, unkept appointments, and cancellations over the previous 12 months was also included. The models were tested using ICHT-specific data from April 2017 to March 2018. Again, including the number of appointments, unkept appointments and cancellations from the previous 12 months... HES data from 2016-17 was used to train the models, then (HES Outpatient data) data from 2017-18 was used for validation."

This would appear to imply that some of the information might be shared between the training and validation datasets. For example, for a given patient, an appointment in March 2017 (with accompanying 12-month prior data) would be employed as training data. However, it would seem that an appointment in say April 2017 by the same patient would then be used in validation, despite it likely sharing similar 12-month prior data (due to eleven of those months overlapping). The conventional arrangement then would then generally be to either stratify by patient, or at the trust level.

The authors might therefore comment on whether such a separation between the training and validation data was achieved, moreover since there might be come confusion as to whether the models were tested only on ICHT-specific data, or the full HES Outpatient data. In general, exactly what models were trained and validated on what data might be more systematically organized (e.g. a "composite prediction (GBM?) model" is suddenly mentioned in the Results section, but apparently not in the previous Methods section, while the GBM Model is only described in a later section after the Results section)

4. The exclusion of cancellations in advance from the analysis would seem to affect the applicability of the results to a real-life implementation, since the trained GBM model would possibly be applied to cases that are eventually cancelled (rather than kept/missed). The authors might wish to discuss whether such cancellations are prevalent, and as such the extent to which they might affect the utility of the models.

5. For the metrics reported under the Gradient Boosting Machine Model section, the Likelihood Ratio and PPV are dependant on patients who were "selected for an intervention". Is this selection based on the 10% sensitivity cut off, and if so, might different cut offs for different specialities be considered given prior knowledge (e.g. 6% unkept appointment rates for medical oncology, compared to 17% for hepatology, from Appendix 2)?

6. There are few details about the GBMs that would allow for some reproducibility. In particular, how were parameters such as the number of trees/shrinkage parameters/bagging fraction chosen? Was there any parameter search involved for each trained model? How was the importance of the various predictors determined? Was there any normalization of the inputs? These details might be appropriately covered in the appendix.

7. The description of "Model sensitivity" mentions that "This is the biggest difference the intervention could possibly make". It is not immediately clear as to what this means.

8. The motivation for using predictive models for targeted interventions might be more extensively analyzed in the Discussion section. In particular, some interventions such as SMS reminders would appear to be feasible for all patients, without the need for a predictive model. 

9. There remain a small number of possible grammatical issues, e.g. "top 10% less likely" -> "top 10% least likely"; references to tables and appendices in the text are generally capitalized.

Reviewer #2: 

This paper is an excellent piece of work. it clearly identifies the issues attributed to unkept appointments and states the research aims very clearly.

The statistical terms have been very well explained as was your methodology.

The tables clearly stated the results, particularly rates and numbers per location and specialisation very succinctly.

I have no criticisms to add of the paper, it is a well written and executed research paper.

Reviewer #3: 

General Feedback- major aspects

Thank you for submitting this manuscript for consideration and I read it with interest as a clinician and researcher who works in this field.

I do not however have any expertise in machine learning and appreciated the description of key terms included.

It would be useful for the reader if you were to explain the change in language from missed to unkept; its not clear, because missed apts have not ever included cancelled apts in the data outputs.

The focus on good population coverage is welcomed and the clear description of the data being used, its benefits and drawbacks. You attempted as best as I was able to discern (not being a stats expert) to account for the data recording variation with respect to missed/unkept appointments. However does this not also mean that some of the most stark data about missingness may have been contained in the top 10%?

 I note that mental health services are not included? In Scottish data they are. Could they all have been in the top 10% or are they not included in HES OP data? I think you need to explain why secondary care mental health services are not included in this data set. I'm making this point because the hospital activity data linked to the GP publshed data you cited does include mental health services and they were the highest generator of repeated missed apts in the Scottish data (currently under review for publication).

I also think that you need to explain in applied terms why exluding the top 10% may impact on the results (even if it may make sense from a machine learning perspective).

Is previous unkept appointments by specialty or any unkept appt, please clarify? Because one potential explanation for high missed appts is patient treatment burden, the patient has so many apts across multiple specialties that they cannot manage to keep them all, which may be why it is linked to cancelled apts too. 

What evidence given the lack of population data on this topic means you can speculate about needs being met elsewhere (line 220 to 231)?

Can you really speculate about rates of unkept appts by English regions when a significant contribution to the data differences may be about data recording and quality if missed appointments are not mandated to be returned? Could this be relevant at specially level too?

I'm uncomfortable with the amount of time spent on possible interventions in the manuscript (250-264)- as the most important finding from this research is that previous missed apts are the strongest predictor of future missingness. As far as I am aware no large or medium scale work has yet evaluated interventions that may work that focus on a history of previous missingness. The evaluations done at any scale do not distinguish between patients who miss one OP apt and those who miss many (and hence probably why studies often contradict each other). This focus on patients at high risk of a pattern of missingness and what works is what is needed- that's my reading of what this important study tells us. And we cannot speculate on what works until research focussed on patients with patterns of high missingness is conducted.

As you are probably aware the evidence that underpins the economic cost per missed apts in the UK is flimsy at best (line 49) and it could be argued have some positive cost benefit (clinicians doing catch up letters etc), so I would suggest including 'estimate' at the very least in your statement about this.

Linked to this would an important recommendation from this work not be that NHS Digital should mandate returning recording of all missed/unkept appointments in line with the need to return all attended ones too? Given that you present a strong case for attention to be paid to this issue.

Minor aspects

Appointment age as a term is misleading- I had to read further down to confirm that it is the age of the patient and not the time elapsed since the appointment was scheduled.

This work was focussed on NHS care in England (not the UK).

In summary though this is an important study that helps distinguish between the patients who miss the occasional apt and those who have patterns of high missed apts within healthcare and it helps the case that interventions for each group are likely to be distinct from each other. My overarching recommendation is a reframing of the paper based on this finding taking the above comments into account is done.

Dr Andrea E Williamson

[LINK]

---

## [Decision Letter · Decision Letter 2]

12 Aug 2021

Dear Dr. Thakrar,

Thank you very much for re-submitting your manuscript "Characterising and reducing the burden of unkept outpatient appointments in the NHS through a national data-driven machine learning approach: A retrospective cohort study" (PMEDICINE-D-20-00544R2) for consideration at PLOS Medicine. We do apologize for the long delay in sending you a decision. 

I have discussed the paper with our academic editor and it was also seen again by two reviewers. I am pleased to tell you that, provided the remaining editorial and production issues are fully dealt with, we expect to be able to accept the paper for publication in the journal.

[LINK]

Please let me know if you have any questions, and we look forward to receiving the revised manuscript.   

Sincerely,

Richard Turner, PhD

rturner@plos.org

Requests from Editors:

Please adapt the title to better match PLOS Medicine style. We suggest: "Characterising the nationwide burden and predictors of unkept outpatient appointments in the NHS in England: A cohort study using a machine-learning approach".

At line 33, please make that "... Data ... were used ...".

At line 39 (abstract), we suggest quoting the estimated proportions contributed by the three predictors mentioned.

Please remove the subsection headed "Key limitations" in your abstract. This material should be located in a new final sentence in the "Methods and findings" subsection, which should begin "Study limitations include ..." or similar and should quote 2-3 of the study's main limitations. 

Please remove the "non-technical summary" following the abstract and instead craft an accessible "Author Summary" section. You may find it help to consult one or two recent research papers in PLOS Medicine to get a sense of the preferred style. 

Please state early in the Methods section whether or not the study had a protocol or prespecified analysis plan.

Please refer to the attached RECORD checklist in the Methods section ("See S1_RECORD_Checklist" or similar). 

Please restructure the start of the Discussion section, as the first paragraph should summarize the study's findings (it appears that this would be achieved if the first two current paragraphs of this section were amalgamated).

Please adapt reference call-outs to precede punctuation throughout the text (e.g., "... up to 6 months [5,6].").

Please remove the information on study funding and competing interests form the end of the main text. This information will appear in the article metadata in the event of publication, via entries in the submission form. 

Noting reference 4, please ensure that all citations have full access details. 

Please use the journal name abbreviation "PLoS ONE" in the reference list.

Please remove all iterations of "[Internet]" from the reference list.

Comments from Reviewers:

*** Reviewer #1: 

We thank the authors for addressing our previous concerns, particularly for the additional Appendix 4 detailing the model specification, and the analysis suggesting that the data leakage did not significantly bias the findings (as illustrated in Figure 3). While the information on RFM models are noted, in principle it would be more ideal to have the training and validation datasets not sharing the same patients. This caveat might be briefly acknowledged if thought appropriate.

*** Reviewer #3: 

Thank you for addressing my feedback comprehensively.

There are some minor typo errors- line 46, 87, 170.

***

[LINK]

---

## [Editor Report · Decision Letter 3]

18 Aug 2021

Dear Dr. Thakrar,

Thank you very much for re-submitting your manuscript "Characterising the nationwide burden and predictors of unkept outpatient appointments in the NHS in England: A cohort study using a machine-learning approach" (PMEDICINE-D-20-00544R3) for consideration at PLOS Medicine.

I have discussed the paper with editorial colleagues, and we will need to ask you to address some further points before we are in a position to proceed further. The remaining issues that should be addressed are listed at the end of this email.

In revising the manuscript for further consideration here, please ensure you address the specific points made by the editors. In your rebuttal letter you should indicate your response to the reviewers' and editors' comments and the changes you have made in the manuscript. Please submit a clean version of the paper as the main article file. A version with changes marked must also be uploaded as a marked up manuscript file.

Please let me know if you have any questions, and we look forward to receiving the revised manuscript.   

Sincerely,

Richard Turner, PhD

rturner@plos.org

Requests from Editors:

At line 26, please make that "appointments cost".

At line 41 in the abstract, immediately prior to the sentence summarizing study limitations, we feel that an additional sentence or two should be added to summarize the inferences from the GBM work beginning at line 205 in the Results. The information about sensitivity may be the most intuitive finding to report in the abstract.

At line 44, please make that "... appointments remain ..." or similar. 

Please revisit the "Author summary", which should consist of three subsections with the following headings, each comprising 3-4 short points (in turn of 1-2 short sentences each):

"Why was this study done?

- 

- 

- 

What did the researchers do and find?

- 

- 

- 

What do these findings mean?

- 

- 

- "

Please use the active voice (e.g., "We investigated ...) in at least one point. 

At line 64, we suggest "impacts".

At line 66, we suggest "... missed more than 2 appointments".

At line 86, please make that "... carried out in a UK setting ...".

At line 90, please make that "potential utility".

At lines 101 and 207, please make that "[data] were used ...".

At line 119, should that be "... appointments and unkept appointments ..."?

At line 187, for example, please avoid using italics for emphasis.

At line 260, please make that "existing literature". 

At line 277, please make that "have been shown".

At line 278, please revisit the wording - perhaps "... the present findings will allow us to go a step farther ..." is intended?

At line 352, please make that "break down" (two words). 

At line 359, please remove the information on competing interests, which will appear in the article metadata via entries in the submission form. 

***

---

## [Editor Report · Decision Letter 4]

25 Aug 2021

Dear Dr Thakrar, 

On behalf of my colleagues and the Academic Editor, Dr Basu, I am pleased to inform you that we have agreed to publish your manuscript "Characterising the nationwide burden and predictors of unkept outpatient appointments in the NHS in England: A cohort study using a machine-learning approach" (PMEDICINE-D-20-00544R4) in PLOS Medicine.

PRESS

Sincerely, 

Richard Turner, PhD 

rturner@plos.org